# Immunogenicity and reactogenicity of a third dose of BNT162b2 vaccine for COVID-19 after a primary regimen with BBIBP-CorV or BNT162b2 vaccines in Lima, Peru

Natalia Vargas-Herrera[1]*, Manuel Fernández-Navarro[1], Nestor E. Cabezudo[2], Percy Soto-Becerra[3], Gilmer Solís-Sánchez[4], Stefan Escobar-Agreda[1], Javier Silva-Valencia[1], Luis Pampa-Espinoza[1], Ricardo Bado-Pérez[5], Lely Solari[1], Roger V. Araujo-Castillo[1]

1 Centro Nacional de Salud Pública, Instituto Nacional de Salud, Lima, Peru, 2 Measles and Rubella National Reference Laboratory, Centro Nacional de Salud Pública, Instituto Nacional de Salud, Lima, Peru, 3 Instituto de Evaluación de Tecnologías en Salud e Investigación (IETSI), Seguro Social de Salud del Peru (ESSALUD), Lima, Peru, 4 Oficina General de Investigación y Transferencia Tecnológica, Instituto Nacional de Salud, Lima, Peru, 5 Centro Nacional de Epidemiología, Prevención y Control de Enfermedades, Ministerio de Salud – Perú, Lima, Peru

* nataliavh84@gmail.com

**Data Availability Statement:** All relevant data are within the paper and its Supporting information file.

## Abstract

### Background

The administration of a third (booster) dose of COVID-19 vaccines in Peru initially employed the BNT162b2 (Pfizer) mRNA vaccine. The national vaccination program started with healthcare workers (HCW) who received BBIBP-CorV (Sinopharm) vaccine as primary regimen and elderly people previously immunized with BNT162b2. This study evaluated the reactogenicity and immunogenicity of the "booster" dose in these two groups in Lima, Peru.

### Methods

We conducted a prospective cohort study, recruiting participants from November to December of 2021 in Lima, Peru. We evaluated immunogenicity and reactogenicity in HCW and elderly patients previously vaccinated with either two doses of BBIBP-CorV (heterologous regimen) or BTN162b2 (homologous regimen). Immunogenicity was measured by anti-SARS-CoV-2 IgG antibody levels immediately before boosting dose and 14 days later. IgG geometric means (GM) and medians were obtained, and modeled using ANCOVA and quantile regressions.

### Results

The GM of IgG levels increased significantly after boosting: from 28.5±5.0 AU/mL up to 486.6±1.2 AU/mL (p<0.001) which corresponds to a 17-fold increase. The heterologous vaccine regimen produced higher GM of post-booster anti-SARS-CoV-2 IgG levels, eliciting a 13% increase in the geometric mean ratio (95%CI: 1.02–1.27) and a median difference of

**Funding:** Yes, this study was funded by the Instituto Nacional de Salud del Peru, study protocol number OI-035-2021.

**Competing interests:** The authors have declared that no competing interests exist.

92.3 AU/ml (95%CI: 24.9–159.7). Both vaccine regimens were safe and well tolerated. Previous COVID-19 infection was also associated with higher pre and post-booster IgG GM levels.

## Conclusion

Although both boosting regimens were highly immunogenic, two doses of BBIBP-CorV boosted with BTN162b2 produced a stronger IgG antibody response than the homologous BNT162b2 regimen in the Peruvian population. Additionally, both regimens were mildly reactogenic and well-tolerated.

## Introduction

The first case of COVID-19 was confirmed in Peru on March 6[th], 2020 [1]. Since then, almost 3,5 million COVID-19 cases and more than 200,000 deaths have been reported [2], making Peru one of the countries with the highest death toll due to COVID-19 in the world [3]. In February 2021, in the midst of a very intense second wave of COVID-19, the healthcare workers (HCW), police and military personnel received the BBIBP-CorV (Sinopharm) inactivated SARS-CoV-2 vaccine [4]. In May 2021, vaccination started on people 60 years and older with the BNT162b2 (Pfizer-BioNTech) mRNA vaccine, and subsequently the vaccination program was extended to the younger population [5] according to vaccine availability, using mainly BBIBP-CorV and ChAdOx1 nCoV-19 (Oxford-AstraZeneca) in under 40s. All regimens consisted of two doses of vaccines, at least 21 days apart.

In October 2021, the Peruvian Ministry of Health (MINSA) approved the administration of third vaccine "booster" doses [6] with BNT162b2. HCWs and people 60 years and older were again prioritized to be vaccinated. With the arrival of the Omicron variant in December 2021, all adults and children above 12 with comorbidities were eligible for the booster dose. Strategies for vaccination against COVID-19 in Peru are permanently being reviewed and updated according to the results of their evaluation. It is important to evaluate the safety and effectiveness of all the vaccination regimens mandated by the MINSA. The aim of our study is to evaluate the reactogenicity and immunogenicity of the third "booster" dose with BNT162b2 in people primarily vaccinated with BBIBP-CorV or BNT16b2 in Lima, Peru.

## Materials and methods

### Design, setting and population

We performed a prospective cohort study in individuals who were administered a BNT162b2 booster dose according to Peruvian COVID-19 vaccination guidelines. The study population included participants aged 18 years and older who had previously received two doses of COVID-19 vaccines 5 to 12 months before. The population comprised two groups: People initially vaccinated with 2 doses of BNT162b2 (Pfizer-Biotech), mainly individuals aged 60 years and older; and people vaccinated with 2 doses of BBIBP-CorV (Sinopharm), mostly HCWs of any profession.

Participants were excluded if, at the time of enrollment, they had active COVID-19 symptoms, allergy to the BNT162b2 vaccine, or reported pregnancy. Participants who had received more than two doses of any COVID-19 vaccine, or received the initial doses abroad were also excluded, as well as participants who did not receive the booster dose within 24 hours after enrollment. Sampling was carried out in a consecutive non-probabilistic manner in four

vaccination centers in Lima that were specifically authorized to administer the booster dose. Sample size was calculated to estimate the geometric mean of the difference between IgG levels before and after the vaccine booster. Considering a difference of IgG levels of 1.09 AU/ml ±1.00 [7], a precision of one tenth of the mean, and a 95% confidence interval, the sample size was 387 subjects. Half that sample size yielded >99% power to test if IgG ratios after boosting were different from 1, including a Bonferroni correction for ten simultaneous comparisons.

## Study procedures

Subjects meeting selection criteria were invited to participate in the study and signed an informed consent form. Clinical and demographic data were registered in a written form, and a 5 ml blood sample was drawn from each participant before receiving the booster dose. Participants were invited for a second visit 14 days after the booster dose and the procedures were repeated.

The main outcome was immunogenicity, assessed through SARS-CoV-2 anti-spike and anti-nucleoprotein IgG antibodies levels. These were measured using the iFlash-SARS-CoV IgG assay (Shenzen YHLO Biotech Co., Ltd, China), a paramagnetic particle chemiluminescent immunoassay (CLIA) using the Immunoassay Analyzer [8]. No lower or upper top values were specified for this assay, although a 10 AU/ml cut-off for positivity point was provided. Test details are provided in S1 Appendix. Sample analysis was performed at the Measles and Rubella National Reference Laboratory of the Instituto Nacional de Salud–Peru.

Other variables analyzed were gender, age group (according to the World Health Organization classification), presence and number of comorbidities (high blood pressure, diabetes mellitus, obesity, asthma, chronic obstructive pulmonary disease, cancer, cardiovascular diseases, others), prior COVID-19 infection (defined as having a prior positive antigenic or molecular test), time in months between the second vaccine dose and the booster dose, time in days between first and second blood sample, and type of primary vaccine regimen (BNT162b2 or BBIBP-CorV).

Safety assessment included self-report of local and systemic adverse reactions (AR) including pain in the injection site, malaise, headache, drowsiness, fever and other events after the BNT162b2 booster dose in both groups. Those were inquired during the two-week follow-up visit. Depending on AR intensity, these were classified as mild or severe according to the Common Terminology Criteria for Adverse Events (CTCAE) [9]. Hospitalizations or deaths until second visit were also recorded.

## Statistical analysis

Categorical variables were described using absolute and relative frequencies, while numerical variables were reported using medians and interquartile ranges (IQR). IgG levels were additionally characterized by geometric means (GM) and geometric standard deviations (GSD). Study variables were compared according to follow-up status, primary vaccine regimen, and adverse reaction presence, using chi-squared and Fisher´s exact test for categorical variables, and Mann-Whitney U test for numerical variables. Only participants that returned for the second visit were included in the reactogenicity and immunogenicity analysis. Crude and Adjusted Poisson regression models with robust standard errors were constructed in order to estimate relative risks (RR) for developing adverse reactions.

For the immunogenicity analysis, comparison between IgG levels before and after vaccine boosting was performed using Wilcoxon Sign Rank test and paired T test for GMs with unequal variances. Bivariate association between the study variables and IgG levels before/after boosting was evaluated two ways: IgG medians were contrasted using Mann-Whitney or

Kruskall Wallis tests, while GMs were compared using Student T or F test for geometric means. In order to model IgG values after vaccine booster, two methods were employed: quantile regression to the median in order to evaluate changes in absolute IgG values; and an ANCOVA approach using IgG geometric means and exponentiated coefficients to evaluate changes in terms of mean fold increase. Robust standard errors were used in both to handle heteroskedasticity of residuals.

All multivariable models were adjusted per age, sex, comorbidity presence, prior COVID-19 infection, time between second and booster dose, vaccine booster regimen, time between first and second serum sample, and IgG levels before booster. The natural logarithm form of the latter was used in an attempt to normalize its distribution. Only in the immunogenicity analysis after booster, continuous numerical variables were modeled using restricted cubic splines in order to handle non-linearity. Spline knots were set according to Harrell's criteria [10]. We demonstrated the adequacy of knots selection through the inspection of partial residual plots and comparing AIC between different spline's parameterizations. All confidence intervals were calculated at 95%, and significant p-values were set at 0.05. All the statistical analyses were performed using Stata v.16 (College Station, TX: StataCorp LLC. 2019).

### Ethical considerations

The study protocol was approved by the National Institute of Health's Institutional ethics committee (approval code: OI-35-21) and all participants signed a voluntary Informed Consent Form.

## Results

### Baseline characteristics

Between November 4 and December 17, 2021, 462 individuals were enrolled. Two participants were excluded for having received their initial vaccine doses abroad (Moderna, mRNA-1273), one for having severe immunosuppression, one who received the initial two doses more than 12 months ago, and one who did not receive the booster dose at all. Of the 457 participants who fulfilled the selection criteria, 285 (62.4%) returned for the second blood sample collection and were eligible for the immunogenicity/reactogenicity analysis (Fig 1). Baseline and demographics characteristics were similar between the group that completed two blood samples and the group lost to follow up (S1 Table).

Patients included had a median age of 46 years (IQR: 36–60) and 190 (66.7%) were female; 214 (75.1%) reported at least one comorbidity and 84 (29.5%) had prior COVID-19 infection. The time between the first two doses ranged between 20 and 71 days, with a median of 21 days. Regarding boosting, time between second and third dose ranged from 5 to 8 full months with a median of 220 days. Median time between first and second blood draw was 15 days (IQR: 14–15). Patients were grouped according to primary vaccine regimen, 56 (19.6%) were primed with BNT162b2 and therefore received a homologous boosting, while 229 (80.4%) were primed with BBIBP-CorV, resulting in a heterologous booster (Table 1). There were some statistically significant differences between both groups. The group primed with BNT162b2 has a median age of 67 compared with 43 in the BBIBP-CorV group (p<0.001); this is the result of the national vaccination program that provided BNT162b2 to elderly population, while BBIBP-CorV was destined to healthcare workers. The time between the first two doses had a median time of 21 days (range:20–33) for the group primed with BNT162b2 and 21 days (range 20–71) for the BBIBP-CorV group. Other differences that arose from this vaccination strategy were differences in sex, since there are more female healthcare workers; comorbidities, since they are more prevalent in elders; and time between the second dose and

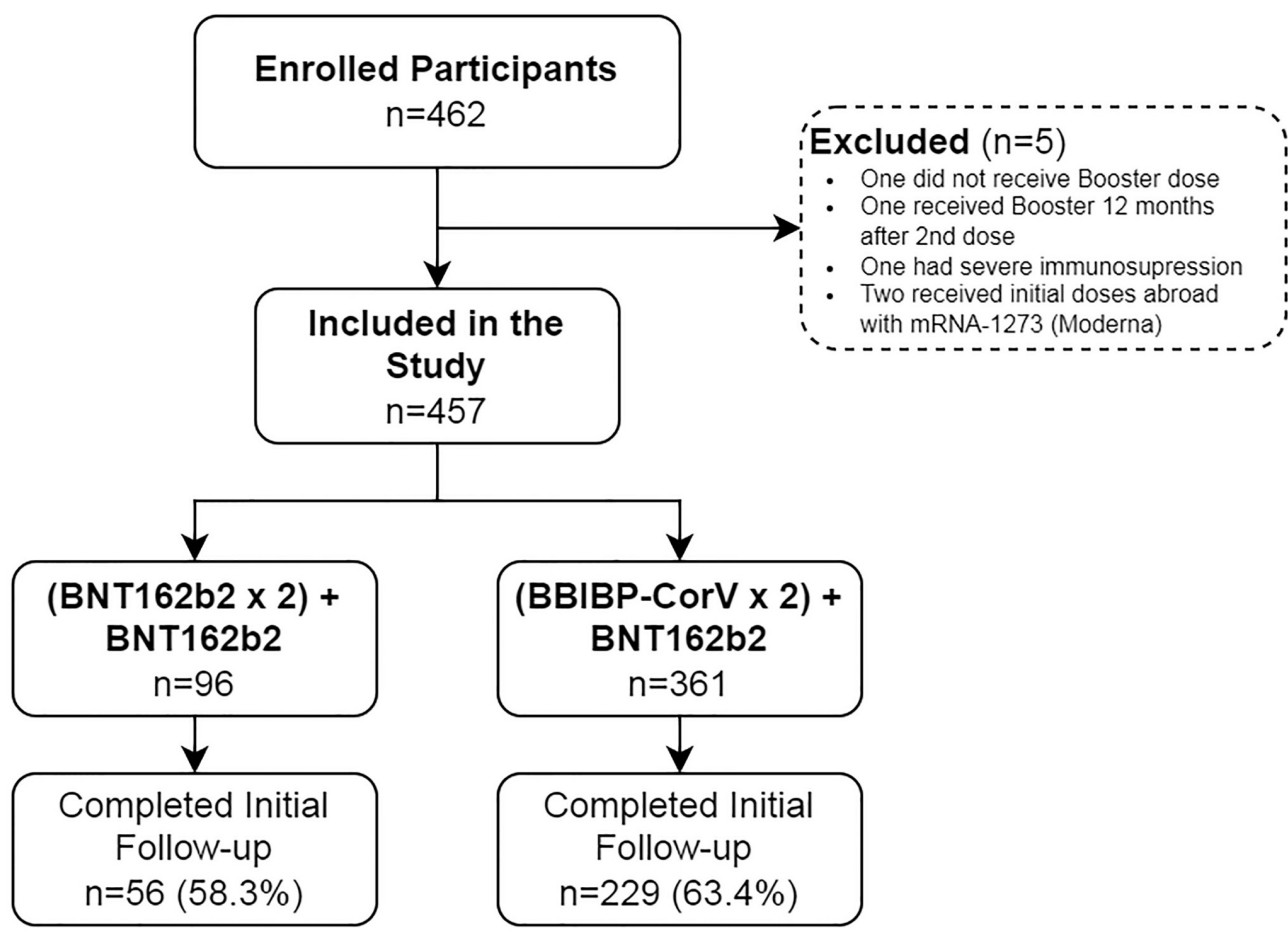

**Fig 1. Participation flowchart.**

booster, since national vaccination started with BBIBP-CorV in healthcare workers, before expanding to people age 60 and older. On the other hand, there were no statistically significant differences by prior COVID-19 infection, or time between blood samples (Table 1).

## Reactogenicity

Among the 285 participants, 251 (88.1%) reported an adverse reaction after booster administration, all of them mild: 244 (85.3%) reported local pain at the injection site, 94 (32.9%) malaise, 79 (27.6%) headache, 43 (15%) drowsiness, 41 (14.3%) fever, and 54 (18.9%) reported other adverse reactions including diarrhea, nausea, vomiting, palpitations, neck/back pain, and one participant reported menstrual cycle changes. In the bivariate analysis, younger age, being female, and having received a heterologous booster were associated with a higher proportion of adverse reactions (S2 Table). In the adjusted regression model, the only characteristic that remained associated was sex: female participants were 13% more likely to develop adverse reactions than male participants (RR 1.12; 95%CI 1.01–1.25) (Table 2).

## Baseline and post-booster immunogenicity

When comparing IgG levels pre versus post booster in the whole group, a marked difference was observed (Fig 2). The GM of IgG levels increased significantly after boosting: from 28.5

**Table 1. Participant characteristics according to primary vaccine regimen (N = 285).**

| | Total | (BNT162b2 x 2) + BNT162b2 | (BBIBP-CorV x 2) + BNT162b2 | p-value* |
|---|---|---|---|---|
| | N = 285 | N = 56 | N = 229 | |
| | n (%) \| Median [IQR] | n (%) \| Median [IQR] | n (%) \| Median [IQR] | |
| **Age (years)** | 46 [36; 60] | 67 [62; 73] | 43 [34; 53] | <0.001‡ |
| **Age Group** | | | | |
| 18–29 years old | 23 (8.1) | 0 (0.0) | 23 (10.0) | <0.001†† |
| 30–59 years old | 189 (66.3) | 6 (10.7) | 183 (79.9) | |
| 60 plus years old | 73 (25.6) | 50 (89.3) | 23 (10.0) | |
| **Gender** | | | | |
| Female | 190 (66.7) | 28 (50.0) | 162 (70.7) | 0.003† |
| Male | 95 (33.3) | 28 (50.0) | 67 (29.3) | |
| **Comorbidity** | | | | |
| No Comorbidities | 214 (75.1) | 28 (50.0) | 186 (81.2) | <0.001† |
| Presence of Comorbidities | 71 (24.9) | 28 (50.0) | 43 (18.8) | |
| **Number of Comorbidities** | | | | |
| No Comorbidities | 214 (75.1) | 28 (50.0) | 186 (81.2) | <0.001† |
| One Comorbidity | 61 (21.4) | 22 (39.3) | 39 (17.0) | |
| Two or more Comorbidities | 10 (3.5) | 6 (10.7) | 4 (1.8) | |
| **List of Comorbidities** | | | | |
| High Blood pressure | 29 (10.2) | 15 (26.8) | 14 (6.1) | <0.001† |
| Diabetes Mellitus | 17 (6.0) | 8 (14.3) | 9 (3.9) | 0.003† |
| Obesity | 7 (2.5) | 0 (0.0) | 7 (3.1) | 0.352†† |
| Asthma/COPD | 12 (4.2) | 2 (3.6) | 10 (4.4) | 1.000†† |
| Cancer (any type) | 5 (1.8) | 3 (5.4) | 2 (0.9) | 0.054†† |
| Cardiovascular Disease | 2 (0.7) | 0 (0.0) | 2 (0.9) | 1.000†† |
| Others | 12 (4.2) | 7 (12.5) | 5 (2.2) | 0.001† |
| **Prior COVID-19 Infection** | | | | |
| No | 201 (70.5) | 45 (80.4) | 156 (68.1) | 0.072† |
| Yes | 84 (29.5) | 11 (19.6) | 73 (31.9) | |
| **Time until booster dose (months)** | | | | |
| 5 | 33 (11.6) | 33 (58.9) | 0 (0.0) | <0.001†† |
| 6 | 78 (27.4) | 20 (35.7) | 58 (25.3) | |
| 7 | 159 (55.8) | 3 (5.4) | 156 (68.1) | |
| 8 | 15 (5.3) | 0 (0.0) | 15 (6.6) | |
| **Adverse Reactions after booster** | | | | |
| No | 34 (11.9) | 13 (23.2) | 21 (9.2) | 0.004† |
| Yes | 251 (88.1) | 43 (76.8) | 208 (90.8) | |
| **Number of Adverse Reactions** | | | | |
| None | 34 (11.9) | 13 (23.2) | 21 (9.2) | <0.001† |
| One | 104 (36.5) | 26 (46.4) | 78 (34.1) | |
| Two or more | 147 (51.6) | 17 (30.4) | 130 (56.8) | |
| **Adverse Reaction occurred** | | | | |
| Local pain | 242 (84.9) | 43 (76.8) | 199 (86.9) | 0.058† |
| Malaise | 93 (32.6) | 11 (19.6) | 82 (35.8) | 0.021† |
| Headache | 79 (27.7) | 6 (10.7) | 73 (31.9) | 0.002† |
| Drowsiness | 43 (15.1) | 3 (5.4) | 40 (17.5) | 0.022†† |
| Fever | 41 (14.4) | 6 (10.7) | 35 (15.3) | 0.382† |
| Others | 54 (19.0) | 1 (1.8) | 53 (23.1) | <0.001†† |

*(Continued)*

**Table 1.** (Continued)

| | Total | (BNT162b2 x 2) + BNT162b2 | (BBIBP-CorV x 2) + BNT162b2 | p-value* |
|---|---|---|---|---|
| | N = 285 | N = 56 | N = 229 | |
| | n (%) \| Median [IQR] | n (%) \| Median [IQR] | n (%) \| Median [IQR] | |
| Time between 1st and 2nd sample (days) | 15 [14; 15] | 14 [14; 17] | 15 [14; 15] | 0.686‡ |

IQR: Interquartile range. IgG: Immunoglobulin G. AU/ml: Arbitrary units per ml.

[†]Chi Square test.

[††]Fisher´s Exact test.

[‡]Mann-Whitney U test.

[*]Comparison between BNT162b2 x 2 + BNT162b2 vs BBIBP-CorV x 2 + BNT162b2.

±5.0 AU/mL up to 486.6±1.2 AU/mL (paired T test: p<0.001) which corresponds to a 17-fold increase. This was also observed for the median: from 29.1 AU/mL (8.4; 93.1) up to 501.9 AU/mL (446.8; 545.4) (Wilcoxon signed rank test: p<0.001).

Regarding COVID-19 baseline humoral status, people aged 60 and older had a higher GM (64.0+/-4.9 AU/ml) than people 18–29 years old (22.9+/-5.4 AU/ml) and 30–59 years old (21.5 +/-4.5 AU/ml). A possible explanation was that elderly were immunized with the BNT162b2 vaccine which has demonstrated to be more immunogenic than BBIBP-CorV according to some studies [11]. However, this trend was reversed for IgG levels after boosting: people 18–29

**Table 2. Regression models using presence of adverse reactions to the vaccine booster as outcome (N = 285).**

| | Crude Models | | Adjusted Model | |
|---|---|---|---|---|
| | RR (95% CI) | p-value* | RR (95% CI) | p-value** |
| **Age Group** | | | | |
| 18–29 years old | 1.26 (1.12; 1.41) | <0.001 | 1.16 (0.97; 1.38) | 0.102 |
| 30–59 years old | 1.13 (1.00; 1.28) | 0.054 | 1.04 (0.88; 1.23) | 0.662 |
| 60 plus years old | Reference | | Reference | |
| **Gender** | | | | |
| Female | 1.15 (1.03; 1.28) | 0.011 | 1.12 (1.01; 1.25) | 0.036 |
| Male | Reference | | Reference | |
| **Comorbidity** | | | | |
| No Comorbidities | Reference | | Reference | |
| Presence of Comorbidities | 0.93 (0.83; 1.04) | 0.189 | 0.97 (0.85; 1.11) | 0.654 |
| **Prior COVID-19 Infection** | | | | |
| No | Reference | | Reference | |
| Yes | 1.08 (0.99; 1.17) | 0.068 | 1.03 (0.94; 1.13) | 0.541 |
| **Time until booster dose (months)** | | | | |
| For each month | 1.03 (0.97; 1.08) | 0.338 | 0.92 (0.85; 1.00) | 0.059 |
| **IgG Titers before booster** | | | | |
| For each natural logarithm | 1.01 (0.95; 1.08) | 0.684 | 1.05 (0.97; 1.13) | 0.203 |
| **Vaccine Booster Regimen** | | | | |
| (BNT162b2 x 2) + BNT162b2 | Reference | | Reference | |
| (BBIBP-CorV x 2) + BNT162b2 | 1.18 (1.02; 1.37) | 0.028 | 1.26 (0.97; 1.63) | 0.079 |

RR: Risk ratio. 95%CI: 95% Confidence Interval. IgG: Immunoglobulin G.

[*] Poisson regression with robust variance, crude models.

[**] Poisson regression with robust variance, adjusted per all listed variables.

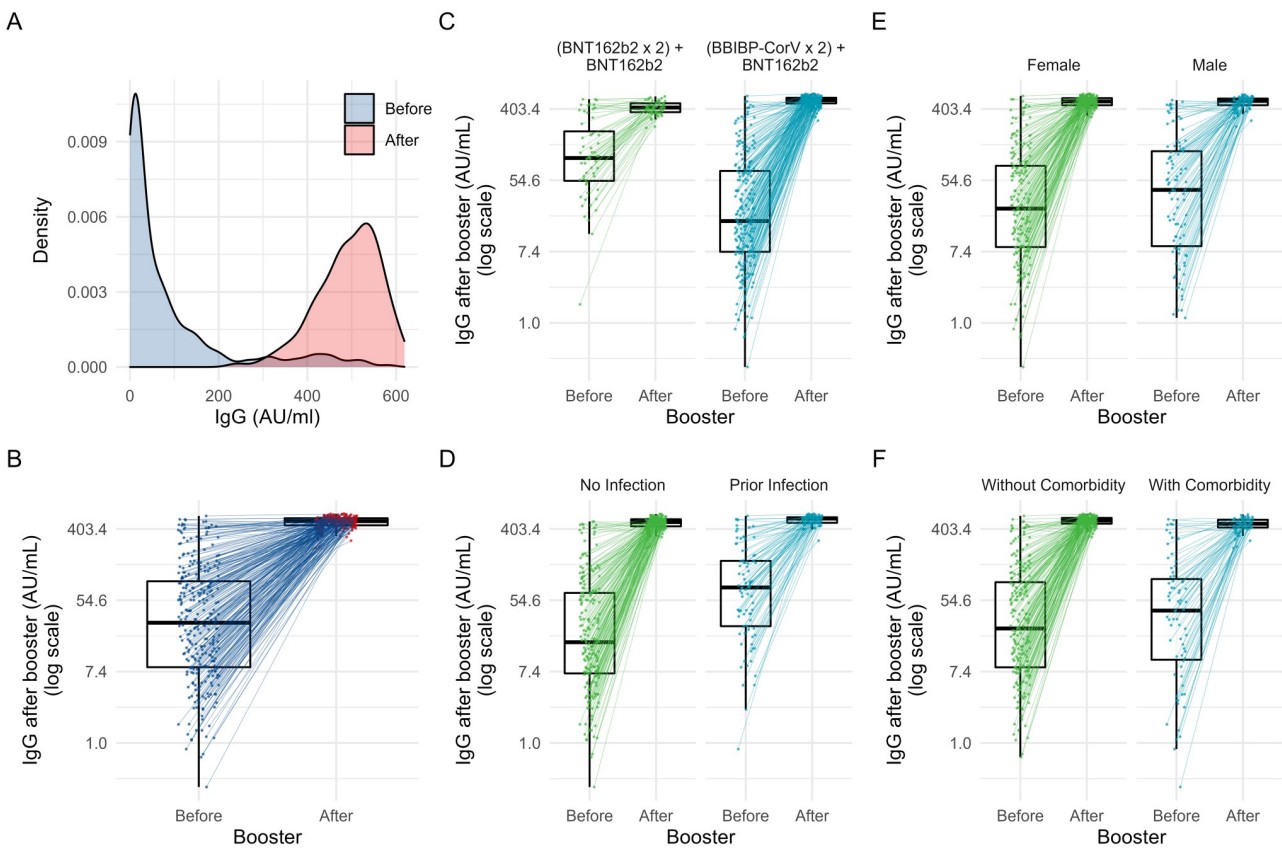

**Fig 2.** A) Density chart showing IgG levels (AU/ml) before and after vaccine booster for the entire sample. B-F) Spaghetti and box plots showing IgG levels (AU/ml) before and after vaccine booster for the entire sample (B), and categorized by vaccine combination (C), by prior COVID-19 infection (D), by sex (E), and by presence of comorbidities (F). Box plots show median as midline, 25 and 75 percentiles as box outer lines, lower and upper adjacent values as line endings, plus outliers.

years old, and 30–59 years old had the higher GMs (518.9+/-1.1 AU/mL and 505.3+/-1.1 AU/mL respectively) compared with people aged 60 years and older (432.3+/-1.2 AU/mL). This trend was also observed when aged was analyzed as a continuous variable (Table 3, Fig 3).

Baseline levels were not different by presence of comorbidities, but after booster levels were lower in people presenting them: 453.8 ±1.2 AU/mL versus 497.9±1.1 AU/mL (p<0.001). A prior COVID-19 infection was associated with a higher GM of baseline levels: 66.0±3.9 AU/mL versus 20.1±4.9 AU/mL (p<0.001), and also with higher post-booster levels: 519.2±1.1 AU/mL versus 473.5±1.2 AU/mL (p<0.001). There were no differences in pre o post booster IgG levels by gender (Table 3, Fig 2). Baseline IgG levels showed a trend towards higher values at shorter periods of time between the second dose and boosting. On the contrary, post-booster IgG levels tend to increase with longer periods of time, except for the 8th month, when IgG levels started to decrease. Regarding time between first and second blood sample, post-booster IgG antibody levels increased sharply until day 15th, then progressively decreased until reaching a steady state (p = 0.003) (Table 3, Fig 3).

The homologous vaccine group had the highest GM of IgG baseline antibody levels when compared to the heterologous vaccine group: 99.5±3.1 AU/mL versus 21.0±4.8 AU/mL (p<0.001). However, this relationship was reversed for post-booster IgG levels: the heterologous vaccine group presented the highest GM when compared to the homologous vaccine group: 505.6±1.1 AU/mL versus 416.0±1.2 AU/mL (p<0.001) (Table 3, Fig 2). Similar

**Table 3. IgG geometric mean titers (AU/ml) before (baseline) and after receiving the COVID-19 vaccine booster dose (N = 285).**

| | Baseline Geometric Mean (GSD) | p-value[*] | After the booster Geometric Mean (GSD) | p-value[*] |
|---|---|---|---|---|
| **Age Group** | | | | |
| 18–29 years old | 22.9 (5.4) | <0.001 | 518.9 (1.1) | <0.001 |
| 30–59 years old | 21.5 (4.5) | | 505.3 (1.1) | |
| 60 plus years old | 64.0 (4.9) | | 432.3 (1.2) | |
| **Gender** | | | | |
| Female | 26.1 (4.8) | 0.200 | 488.8 (1.2) | 0.511 |
| Male | 34.1 (5.3) | | 482.2 (1.2) | |
| **Comorbidity** | | | | |
| No Comorbidities | 27.7 (4.8) | 0.629 | 497.9 (1.1) | <0.001 |
| Presence of Comorbidities | 31.1 (5.9) | | 453.8 (1.2) | |
| **Number of Comorbidities** | | | | |
| No comorbidities | 27.7 (4.8) | 0.455 | 497.9 (1.1) | <0.001 |
| One comorbidity | 28.5 (5.6) | | 456.7 (1.2) | |
| Two or more comorbidities | 53.5 (8.2) | | 436.2 (1.2) | |
| **Prior COVID-19 infection** | | | | |
| No Infection | 20.1 (4.9) | <0.001 | 473.5 (1.2) | <0.001 |
| Prior Infection | 66.0 (3.9) | | 519.2 (1.1) | |
| **Time until booster dose (months)** | | | | |
| 5 | 112.6 (2.9) | <0.001 | 424.3 (1.2) | |
| 6 | 29.7 (4.7) | | 463.4 (1.2) | |
| 7 | 19.8 (4.7) | | 511.4 (1.1) | <0.001 |
| 8 | 55.2 (6.5) | | 499.7 (1.1) | |
| **Vaccine Booster Regimen** | | | | |
| (BNT162b2 x 2) + BNT162b2 | 99.5 (3.1) | <0.001 | 416.0 (1.2) | <0.001 |
| (BBIBP-CorV x 2) + BNT162b2 | 21.0 (4.8) | | 505.6 (1.1) | |

IgG: Immunoglobulin G. AU/ml: Arbitrary units per ml. GSD: geometric standard deviation.

[*] Student T or F test for geometric means.

associations were observed when comparisons were stratified by booster regimen (S3 Table), or when performed using medians and IQRs (S4 Table).

Based on these results, two multivariable models were constructed. Both, the ANCOVA and the quantile regression models showed that prior COVID-19 infection was associated with higher post booster levels with a 6% increase in the geometric mean ratio (95%CI: 1.02–1.10) and a median difference of 29.1 AU/ml (95%CI: 11.5–46.7). BBIBP-CorV priming was also associated with higher post booster IgG levels, eliciting a 13% increase in the geometric mean ratio (95%CI: 1.02–1.27) and a median difference of 92.3 AU/ml (95%CI: 24.9–159.7) (Table 4). Regarding the non-linear terms of both regression models, the only significant correlation was between higher IgG levels before booster with higher levels post-booster, as seen in Fig 4. Associations with age, gender, comorbidities, time until booster, and time until second sample disappeared after adjustment. Individual coefficients for each spline of the non-linear terms are shown in the S5 Table for the quantile regression model, and in the S6 Table for the linear regression model.

## Discussion

In this prospective cohort study, we report the humoral immunogenicity of a BNT162b2 vaccine booster in persons having been primarily vaccinated with either two doses of

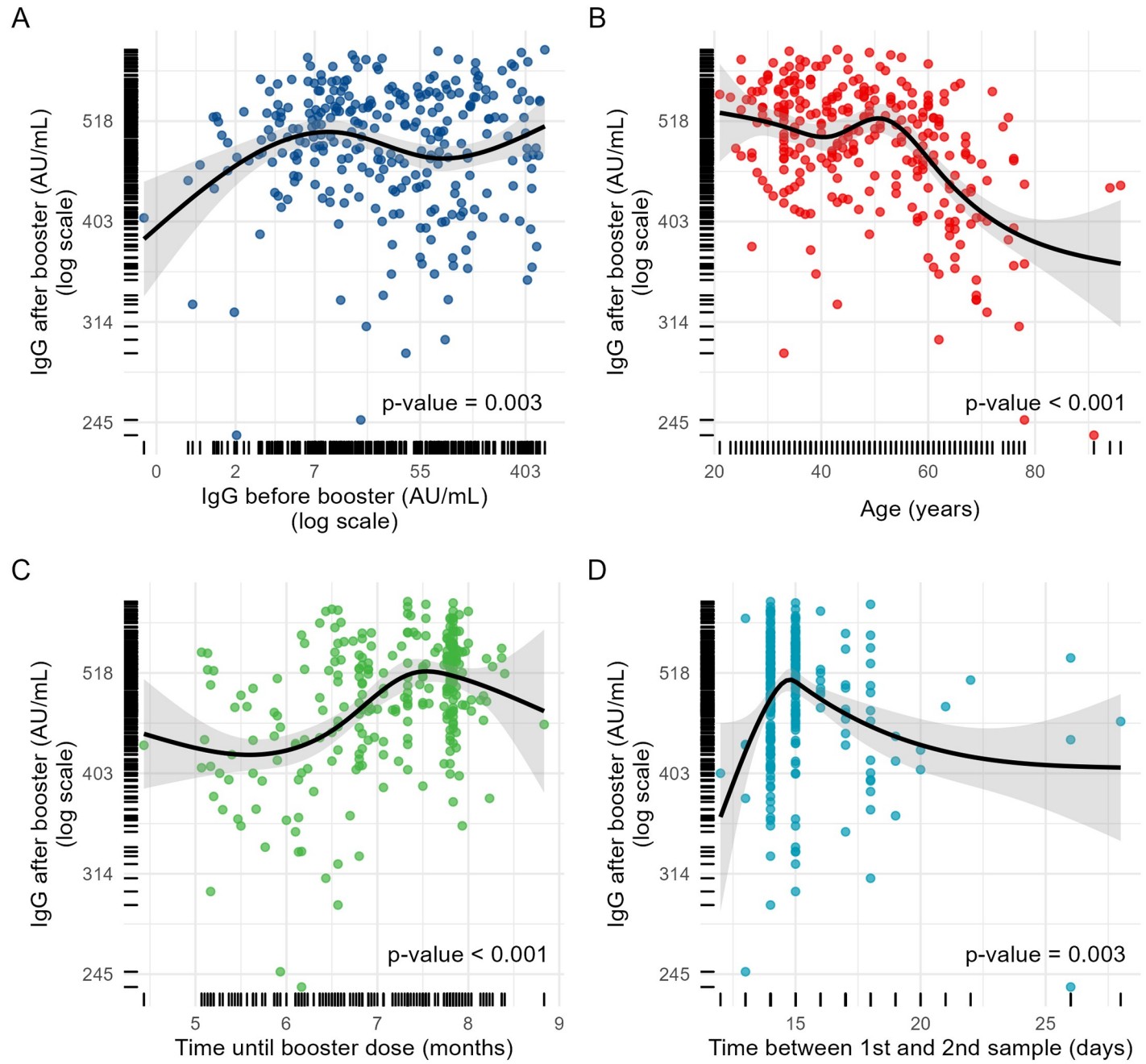

**Fig 3. Bivariate scatter plots plus linear fit lines with 95% confidence intervals.** IgG levels (AU/ml) after vaccine booster (logarithm scale) are shown on the y-axis of all graphics. Numeric variables in the x-axis are displayed using restricted cubic B-splines functions with the spline knots set according to Harrell's criteria. The x-axis displays: IgG levels before booster in a logarithm scale (A), age in years (B), time between second the third vaccine dose in months (C), and time between first and second blood sample in days (D).

BBIBP-CorV or BNT162b, as well as the reactogenicity produced. To our knowledge, this is the first study comparing immunogenicity of these regimens in Latin America. Noteworthy, baseline antibody levels were not uniformly distributed, and participants with prior COVID-19 had significantly higher levels before boosting. Interestingly, baseline levels were higher for people primed with the BNT162b2 vaccine, although people who received BBIBP-CorV as

**Table 4. Adjusted regression models using IgG levels (AU/ml) after vaccine booster as outcome (N = 285).**

| | Multivariable Linear Regression | | Multivariable Quantile Regression | |
|---|---|---|---|---|
| | GMR (95% CI) | p-value[a] | MD (95% CI) | p-value[a] |
| **Age (years)** | | | | |
| (Non-linear term)[b] | * | 0.585[c] | * | 0.080 |
| **Gender** | | | | |
| Female | Reference | | Reference | |
| Male | 1.01 (0.97; 1.04) | 0.650 | 5.39 (-7.68; 18.46) | 0.417 |
| **Comorbidity** | | | | |
| No Comorbidities | Reference | | Reference | |
| Presence of Comorbidities | 0.98 (0.94; 1.03) | 0.460 | 1.88 (-14.24; 17.99) | 0.819 |
| **Prior COVID-19 Infection** | | | | |
| No | Reference | | Reference | |
| Yes | 1.06 (1.02; 1.10) | 0.004 | 29.11 (11.49; 46.73) | 0.001 |
| **Time until booster dose (days)** | | | | |
| (Non-linear term)[b] | * | 0.084[c] | * | 0.281 |
| **Vaccine Booster Regimen** | | | | |
| (BNT162b2 x 2) + BNT162b2 | Reference | | Reference | |
| (BBIBP-CorV x 2) + BNT162b2 | 1.13 (1.01; 1.27) | 0.041 | 92.3 (24.90; 159.7) | 0.007 |
| **Time between 1st and 2nd sample** | | | | |
| (Non-linear term)[b] | * | 0.055[c] | * | 0.305 |
| **Natural Log of IgG titers before Booster** | | | | |
| (Non-linear term)[b] | * | <0.001[c] | * | 0.003 |

IgG: Immunoglobulin G. AU/ml: Arbitrary units per ml. GMR: Adjusted Geometric Mean Ratio. MD: Adjusted Median Difference. 95%CI: 95% Confidence Interval.

[a] All p-values were obtained using a robust standard error estimator to address heteroskedasticity.

[b] The non-linear effect of age, time until booster dose, time between 1st and 2nd sample and natural log of IgG titers before booster in multivariable linear regression are shown in Fig 4.

[c] p-values for multiple coefficients of B-splines basis functions were tested using a heteroskedasticity version of F-Statistic for a joint hypothesis testing.

* Details about coefficients for B-splines are show in S2 and S3 Tables.

primary regimen have been vaccinated for a longer period of time, and it was expected that their IgG levels will be lower at the moment of boosting. In addition, the lower antibody levels could also be expected because of the overall lower antibody immunogenicity of the BBIBP-CorV vaccine compared to BNT162b2 [11].

Our results show that the administration of a BNT162b2 booster significantly elicited robust humoral responses measured by IgG titers in all the different groups studied, regardless of their baseline levels or primary regimen received. This phenomenon has been well-described, even for people primed with inactivated vaccines such as BBIBP-CorV. For instance, a Peruvian one-arm study reported a strong humoral response after a heterologous BNT162b2 booster in HCWs primed with the inactivated BBIBP-CorV vaccine [12], even higher than the 17-fold increase found in our study. In Lebanon, a prospective cohort study comparing a BNT162b2 booster versus no booster in BBIBP-CorV vaccinated people, found that boosting elicited higher anti-spike IgG geometric mean titers: 8040 BAU/mL (95%CI: 4612–14016) versus 1384 BAU/mL (95%CI: 1063–1801) p<0.001 [13]. However, none of these studies included more than one vaccine regime.

In our study, we found that the heterologous combination was more immunogenic than the homologous one, after adjustment by age, gender, comorbidities, prior COVID-19

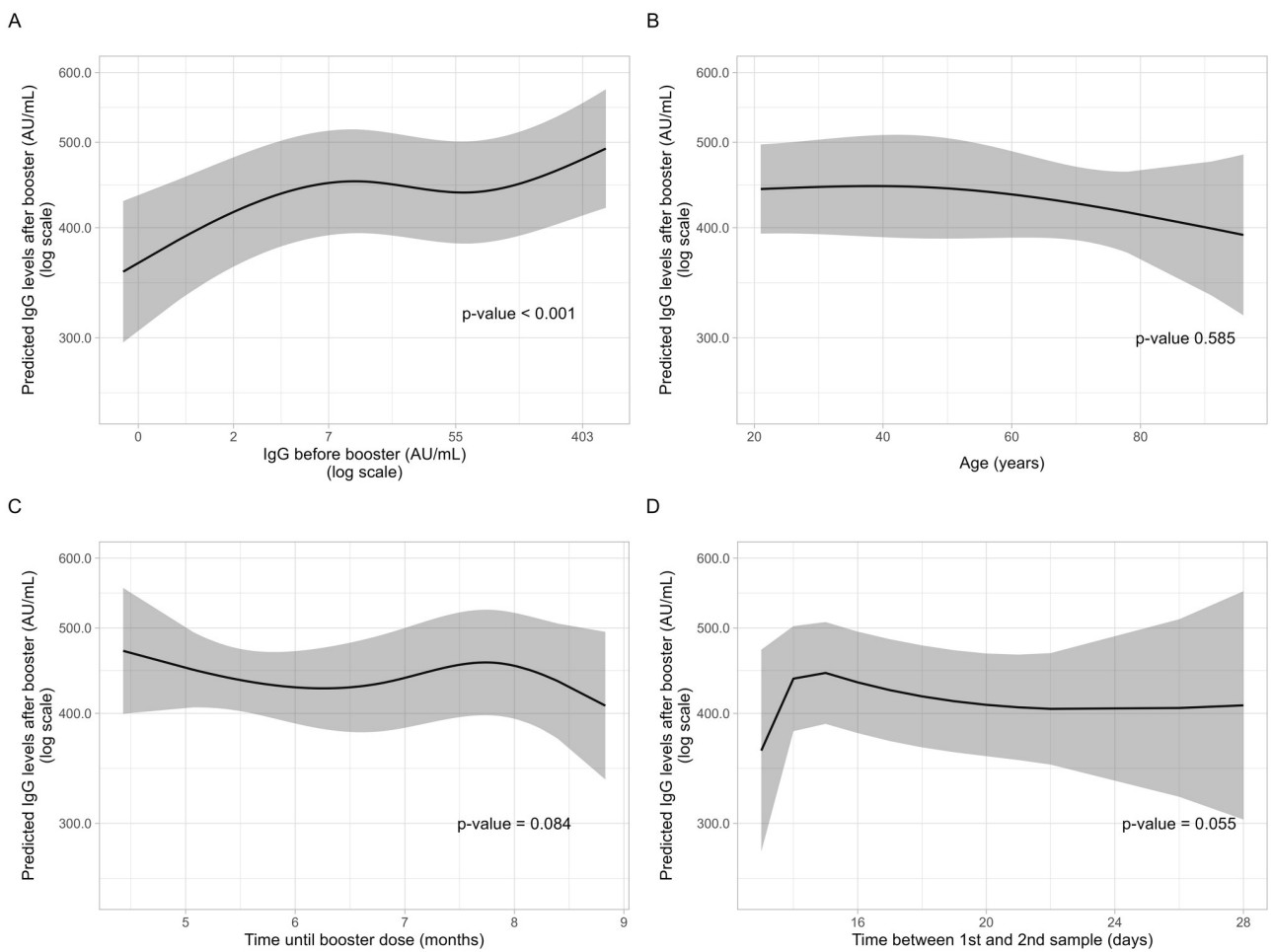

**Fig 4. Predicted IgG levels (AU/ml) after vaccine booster (logarithm scale) with 95% confidence intervals obtained from a multivariate linear model using geometric means and robust standard errors (y-axis).** Numeric variables in the x-axis were treated as restricted cubic B-splines functions with the spline knots set according to Harrell's criteria. The x-axis displays IgG levels before booster in a logarithm scale (A), age in years (B), time between second the third vaccine dose in months (C), and time between first and second blood sample in days (D).

infection, time until booster, time between samples, and baseline IgG levels. The phenomenon of higher humoral response after a heterologous booster has been described in previous studies assessing different COVID-19 vaccines [14–17]. It has also been reported in combinations containing other inactivated virus vaccines. In Chile, Vargas et al. found that, in people primed with CoronaVac (Sinovac), a heterologous booster with BNT162b2 or ChAdOx1 increased anti-spike IgG antibody titers more strongly than the corresponding homologous booster [14].

The use of heterologous vaccine regimens for the second dose or as booster has been investigated before the SARS-CoV-2 pandemic for infectious diseases such as HIV, HPV, influenza, malaria and Ebola [18, 19]. In the COVID-19 pandemic context, both animal and human studies that mixed adenovirus and mRNA vaccines, in general showed higher antibody and T-cell responses when compared to 2 doses of the same vaccine platform [20–22]. The possible mechanism for the higher immune responses when using different vaccine platforms could be explained by evoking different immune pathways which produces stronger and longer-lasting T-cell and B-cell (both IgG and neutralizing antibodies) responses [18]. In the particular case of inactivated COVID-19 vaccines such as BBIBP-CorV, as these contain additional SARS--CoV-2 proteins such as the nucleoprotein, could produce a wider immunological response

than produced by the spike protein. This mechanism could also reduce the immune escape of SARS-CoV-2 variants [15]. This potential advantage could be potentiated by combining an inactivated virus vaccine with an mRNA vaccine, since this last one was the most immunogenic in the COV-BOOST clinical trial when used as part of a heterologous booster regimen [23].

Due to the expected waning effectiveness over time of COVID-19 vaccines, a third dose has demonstrated to increase protection against SARS-CoV-2 infection, severe disease and death [24] which is extremely important in a context of very transmissible variants such as Omicron (B.1.1.529) and its descendant lineages BA.1 and BA.2.

We also observed that participants with prior COVID-19 infection had higher IgG antibody titers post booster, a finding that has been described in studies assessing immunogenicity in vaccinated people with and without previous COVID-19 infection [11]. This is explained by hybrid immunity to SARS-CoV-2 (when vaccine-generated immunity is combined with natural immunity), which induces a potent immune response that can result in 25 to 100 times higher antibody levels due to CD4+T and memory B cells [25].

Regarding reactogenicity, our findings showed that despite most participants reporting at least one adverse event, all of these were mild, and without significant differences between the homologous and heterologous vaccine regimens. In addition, we found that more female participants developed adverse reactions than males, which has been previously described and may be explained by the fact that women are known to elicit stronger innate and adaptive immune responses to foreign antigens than men [26].

Some limitations in our study ought to be acknowledged. In the first place, all the participants were enrolled in vaccination centers from Lima through a non-probabilistic sampling, which could affect the representativeness of the general boosted population in Peru. Secondly, there was an important percentage of loss of follow-up, with almost a third of the enrolled participants not returning on time for their second blood sample. However, the sample size was still enough for a multivariate comparison of IgG levels pre/post booster, and there were no statistically significant differences between the people who completed the second visit and those who did not. An additional problem was the varying time between first and second IgG measurements; although the indication was to return 14 days +/-48 hours after boosting, a significant number came later, up to 28 days after boosting. A further limitation is the under-reporting of prior SARS-CoV-2 due to presence of asymptomatic infections with no testing. Another limitation is the use of a dual-reactive assay (reactivity against the spike and the nucleoprotein) to measure IgG levels in our study, given that the BBIBP-CorV inactivated whole virus vaccine could have induced antibodies against both proteins while BNT162b2, while mRNA vaccines exclusively induce antibodies against the spike. Another important limitation is that due to the vaccine program rolled up in Peru, there were pronounced differences between participants characteristics vaccinated with the homologous and the heterologous regimen, particularly the median age, however, we used robust adjustment strategies for observational studies. Finally, although we measured humoral response by the overall binding reactivity, we did not include neutralizing antibodies or cellular immunity response, although binding antibody titers have been found to correlate with protective efficacy [27].

On the other hand, one of the main strengths of our study is that we included a relatively large number of participants with different ages that were closely followed over time and thus the data obtained regarding immunogenicity and reactogenicity is reliable. We also had a relatable form of measuring prior COVID-19 infection and time of initial vaccines using the Peruvian Ministry of health datasets. Finally, we were extremely careful modeling the IgG levels after boosting using geometric means ratios for the outcome, and applying restricted splines for non-linear numeric exposures. The relevance of this study is mainly related to the

information it offers about the BBIBP-CorV vaccine and combinations of it, for which there is scarcity of evaluation studies. For Peru, the availability of this vaccine for prioritized population such as HCW was very important in moments when other platforms, such as mRNA vaccines, were only available in few countries. Confirming that people receiving it as a primary regime are probably very well protected against subsequent infections with subsequent vaccine doses of other vaccines, now widely available, is reassuring.

In conclusion, two doses of BBIBP-CorV boosted with one BNT162b2 dose elicited very high IgG antibody responses, and three BNT162b2 doses induced a similar response. Both regimens were safe and well tolerated. In addition, the antibody titers rising trend after the third vaccine dose in our study indicates that subsequent boosters could be spaced and prioritized in certain populations such as elderly and immunosuppressed. This reaffirms the importance of mix-and-match strategies that also include inactivated vaccines in order to overcome vaccine availability obstacles.

## Supporting information

**S1 Appendix. IgG COVID-19 antibody assay description.**
(DOCX)

**S1 Table. Participants characteristics according to follow-up status (N = 457).**
(DOCX)

**S2 Table. Characteristics of participant with complete follow-up according to presence of adverse reactions to the vaccine booster (N = 285).**
(DOCX)

**S3 Table. IgG geometric mean titers (AU/ml) before (baseline) and after receiving the COVID-19 vaccine booster dose stratified by booster regimen (N = 285).**
(DOCX)

**S4 Table. IgG median titers (AU/ml) before (baseline) and after receiving the COVID-19 vaccine booster dose (N = 285).**
(DOCX)

**S5 Table. Adjusted quantile regression model using IgG levels (AU/ml) after vaccine booster as outcome showing coefficients for each spline (N = 285).**
(DOCX)

**S6 Table. Adjusted linear regression model using IgG levels (AU/ml) after vaccine booster as outcome showing coefficients for each spline (N = 285).**
(DOCX)

**S1 Data.**
(XLSX)

## Acknowledgments

The authors are grateful to all participating patients and their families, the field research team in charge of drawing the blood samples and the laboratory team in charge of processing the samples.

## Author Contributions

**Conceptualization:** Manuel Fernández-Navarro, Lely Solari.

**Data curation:** Manuel Fernández-Navarro, Nestor E. Cabezudo, Percy Soto-Becerra, Gilmer Solís-Sánchez, Roger V. Araujo-Castillo.

**Formal analysis:** Natalia Vargas-Herrera, Percy Soto-Becerra, Gilmer Solís-Sánchez.

**Investigation:** Natalia Vargas-Herrera, Stefan Escobar-Agreda, Javier Silva-Valencia, Luis Pampa-Espinoza, Ricardo Bado-Pérez, Lely Solari, Roger V. Araujo-Castillo.

**Methodology:** Nestor E. Cabezudo, Javier Silva-Valencia.

**Project administration:** Natalia Vargas-Herrera, Nestor E. Cabezudo.

**Resources:** Nestor E. Cabezudo.

**Software:** Gilmer Solís-Sánchez.

**Supervision:** Natalia Vargas-Herrera, Manuel Fernández-Navarro, Lely Solari, Roger V. Araujo-Castillo.

**Visualization:** Gilmer Solís-Sánchez.

**Writing – original draft:** Natalia Vargas-Herrera, Manuel Fernández-Navarro, Percy Soto-Becerra, Stefan Escobar-Agreda, Javier Silva-Valencia, Luis Pampa-Espinoza, Ricardo Bado-Pérez, Roger V. Araujo-Castillo.

**Writing – review & editing:** Natalia Vargas-Herrera, Manuel Fernández-Navarro, Gilmer Solís-Sánchez, Stefan Escobar-Agreda, Javier Silva-Valencia, Ricardo Bado-Pérez, Lely Solari, Roger V. Araujo-Castillo.

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
