## [Decision Letter · Decision Letter 0]

28 Jun 2022

PONE-D-22-12508Immunogenicity and reactogenicity of a third dose of BNT162b2 vaccine for COVID-19 after a primary regimen with BBIBP-CorV or BNT162b2 vaccines in Lima, Peru.PLOS ONE

Dear Dr. Vargas Herrera,

Thank you for submitting your manuscript to PLOS ONE. After careful consideration, we feel that it has merit but does not fully meet PLOS ONE’s publication criteria as it currently stands. Therefore, we invite you to submit a revised version of the manuscript that addresses all the points raised during the review process.

We look forward to receiving your revised manuscript.

Kind regards,

Daniela Flavia Hozbor

Academic Editor

PLOS ONE

Journal Requirements:

 "Yes, this study was funded by the Instituto Nacional de Salud del Peru, study protocol number OI-035-2021." 

Reviewers' comments:

Reviewer's Responses to Questions

**Comments to the Author**

1. Is the manuscript technically sound, and do the data support the conclusions?

Reviewer #1: Partly

Reviewer #2: Partly

2. Has the statistical analysis been performed appropriately and rigorously? 

Reviewer #1: Yes

Reviewer #2: Yes

3. Have the authors made all data underlying the findings in their manuscript fully available?

Reviewer #1: No

Reviewer #2: Yes

4. Is the manuscript presented in an intelligible fashion and written in standard English?

Reviewer #1: Yes

Reviewer #2: No

5. Review Comments to the Author

Reviewer #1: In this manuscript, Vargas-Herrera and colleagues investigate the response to a BNT162b2 booster vaccination in elderly individuals previously immunized with BNT162b2 as well as younger health care workers previously vaccinated with the inactivated BBIBP-CorV vaccine. To this end, reactogenicity was assessed and SARS-CoV-2-reactive IgG levels were determined on the day of and 14 days after the booster dose.

The authors demonstrate that an mRNA-based booster dose is generally well tolerated and that it induces a strong increase in SARS-CoV-2-reactive IgG levels. This effect is more pronounced in individuals that received a heterologous vaccination regimen.

The results of this study recapitulate the findings of numerous earlier studies, both on tolerability and humoral immunity, and extends them to cohorts from Peru. The results are overall plausible and in line with the results of previous studies. Most of the limitations of the study are acknowledged by the authors. Because the BBIBP-CorV vaccine is widely used but relatively little studied compared to other vaccines, analyzing the effects of booster vaccinations after a primary BBIBP-CorV dose is important.

There are several aspects that I would suggest the authors to address in a revised version of the manuscript.

Major comments:

1)

IgG levels are determined using an assay that tests for reactivity against both the spike protein and the nucleoprotein of SARS-CoV-2 without differentiating this reactivity. Because the BBIBIP-CorV vaccine is an inactivated whole virus vaccine, it will induce antibodies against both proteins (spike and nucleoprotein). In contrast, the BNT162b2 mRNA vaccine exclusively induces antibodies against the spike. This complicates comparisons of the IgG levels as the assay used will pick up two classes of antibodies after BBIBP-CorV vaccination but only class after BNT162b2 vaccination. Ideally, samples would be analyzed using an IgG assay that determines only spike reactivity. Alternatively, could the contribution of anti-N antibodies to the determined signal be analyzed for a representative subset of samples? At a minimum, the use of a dual-reactive assay should be discussed as a limitation for the interpretation of this study.

2)

Immunogenicity results are provided as AU/ml (arbitrary units/ml). To facilitate comparison to results of other trials, it would be helpful if a conversion factor to the international standard BAU/ml using the WHO reference sample with this assay could be provided.

3)

Lines 224-226: Numbers in the text (RR 1.13, 95% CI 1.02-1.26) do not match numbers in Table 2 (female gender, adjusted model; RR 1.12, 95% CI 1.01-1.25)

4)

Lines 243-245/Table 3: The higher baseline titers in older individuals are not unexpected as the majority of elderly individuals had received primary immunizations with the much more immunogenic BNT162b2 vaccine. This should be pointed out here. Because the BNT162b2 is overall more immunogenic than the BBIBP-CorV vaccine, it would be helpful to have Table 3 (or an additional table) be split up into the different primary vaccines as well.

5)

Lines 338-342: While protection induced by inactivated virions could potentially be wider, the statement in its current form is a bit misleading as the lower protection of inactivated virion vaccines compared to the highly immunogenic mRNA vaccines seems well established.

6)

Lines 349-351: The authors state that antibody titers before the booster inversely correlated with the post-boost titers, suggesting that shorter boosting intervals may not be beneficial. In this study, this observation is very likely to be confounded by the fact that young individuals, which generally respond to the BNT162b2 vaccine with higher IgG titers than older individuals, had received the much less immunogenic BBIBP-CorV vaccine first. This results in the relatively strong increase in younger individuals (poorly immunogenic first vaccine, very strong response to booster). In contrast, older individuals had higher baseline titers due to the more immunogenic BNT162b2 vaccine compared to BBIBP-CorV (although titers in younger individuals receiving BNT162b2 would have been even higher), and then responded less strongly to the BNT162b2 booster dose than younger individuals.

Additional comments

7)

Table 1: While i.) the elapsed time between the second vaccine dose and the booster dose as well as ii.) the time between the the booster dose and the follow-up visit are described, information of the time between the first and second dose is missing (line 86 says: at least 21 days apart). As prolonged vaccination intervals are known to affect the humoral vaccine response, it would be helpful for the interpretation of the results and comparison of the groups to also include this measure.

8)

Table 1: Make clear within the table what groups the p values are comparing

9)

Figure 2: Instead of the natural logarithm, it appears advisable to display the data in the much more commonly used log10 format, which will help with comparison to other results. In addition, the legend needs to define lines, boxes and whiskers.

10)

Figure 3: The 95% confidence intervals are a bit difficult to make out and it would be helpful to display them in a more prominent shade of grey.

11)

Line 65: “13% fold” does not make sense

12)

Line 67: Make clear what “both” is referring to

13)

Line 144: Typo, should be BNT162b2

14)

Line 144-145: Beyond “mild” and “severe”, CTCAE also has a “moderate” classification. Was it deliberately excluded?

15)

Line 201: Ranged would be a better word than oscillated

16)

Lines 212-213: Better to write that there were “no statistically significant” differences than just “no” differences.

17)

Line 221: Better: nausea than nauseas

18)

Line 309: The lower antibody levels could also be expected because of the overall lower immunogenicity of the BBIBP-CorV vaccine compared to BNT162b2.

19)

Line 331-332: Heterologous vaccines are common ‘practice’ for several of the diseases (e.g., HIV, HPV). ‘Has been investigated’ might therefore be a better choice of words than ‘has been practiced’.

20)

Line 376: While the authors rightfully acknowledge the lack of analyses on neutralizing serum titers as a limitation of the study, the term ‘broadly’ for the description of the extend of humoral response measurements seems to wide to me (it is a rather narrow measurement of the humoral response). One could rather write “overall binding reactivity” or something similar.

21)

Line 395: “In addition, the antibody titers rising trend after the second vaccine dose in our study indicates that subsequent boosters could be spaced and prioritized in certain populations such as elderly and immunosuppressed.” I am not entirely sure what this statement is referring to. Trends after the second vaccine dose (prior to the booster) were not investigated. Should this refer to the higher titers after the booster dose compared to after the second vaccine dose?

Reviewer #2: Thanks for the opportunity to review.

I have a few comments regarding the manuscript that should be addressed.

- Overall the manuscript is written with a reasonable standard of english but could do with a thorough grammar edit.

The study is interesting, but it is difficult to draw any major conclusions about the impact of booster vaccines for the following reasons:

- It is expected that there would be such an increase in antibody titres early after a booster dose, 2 weeks is too early to assess immunogenicity and durability of response and a later sample would be more meaningful in response interpretation.

- Only IgG was measured, not neutralising antibody which may be more accurate in interpreting immune response correlating with protection

- In view of this is is preferable to tone these findings down as unfortunately it may be misleading to say that these responses result in protection and to correlate them to protection. The findings are interesting, but need to be focussed on what can be derived, rather than making assumptions

- There are also substantial differences between the homologous and heterologous groups and a comparison is very difficult even with some parameters controlled for

In addition:

- How was fever assessed?

-Was any formal measurement of local reactogenicity conducted (measure size of swelling, erythema etc.) Was FDA guidance used to evaluate this?

- How was prior COVID-19 assessed, given that it is expected that by Dec 21 close to 80% of the global population had been infected with COVID-19 as evidenced by antibody testing. Even asymptomatic infection would give an Ab response and this is likely to confound baseline data.

- In line 225 heterologous has been duplicated- I'm assuming that homologous is the correct term

There are significant differences

6. PLOS authors have the option to publish the peer review history of their article (what does this mean?). If published, this will include your full peer review and any attached files.

Reviewer #1: No

Reviewer #2: No

---

## [Author Response · Author response to Decision Letter 0]

23 Aug 2022

Manuscript reference number: PONE-D-22-12508

Title: Immunogenicity and reactogenicity of a third dose of BNT162b2 vaccine for COVID-19 after a primary regimen with BBIBP-CorV or BNT162b2 vaccines in Lima, Peru.

Dear Emily Chenette

Senior Editor,

Plos One

We thank the editor and the two reviewers for their comments on our manuscript, below is our response to each point. We are uploading a version of the manuscript with track changes, and a clean version with all the changes accepted. We hope that we satisfyingly addressed all the comments and that the manuscript will be now suited for publication 

Best Regards,

Natalia Vargas, MD

Instituto Nacional de Salud (INS)

Lima, Peru

August 16th 2022

AUTHOR´S RESPONSE TO COMMENTS:

Reviewer #1: 

• IgG levels are determined using an assay that tests for reactivity against both the spike protein and the nucleoprotein of SARS-CoV-2 without differentiating this reactivity. Because the BBIBP-CorV vaccine is an inactivated whole virus vaccine, it will induce antibodies against both proteins (spike and nucleoprotein). In contrast, the BNT162b2 mRNA vaccine exclusively induces antibodies against the spike. This complicates comparisons of the IgG levels as the assay used will pick up two classes of antibodies after BBIBP-CorV vaccination but only class after BNT162b2 vaccination. Ideally, samples would be analysed using an IgG assay that determines only spike reactivity. Alternatively, could the contribution of anti-N antibodies to the determined signal be analysed for a representative subset of samples? At a minimum, the use of a dual-reactive assay should be discussed as a limitation for the interpretation of this study.

RE: Thank you for your comment, we have added that limitation in lines 381-385 as suggested.

• Immunogenicity results are provided as AU/ml (arbitrary units/ml). To facilitate comparison to results of other trials, it would be helpful if a conversion factor to the international standard BAU/ml using the WHO reference sample with this assay could be provided.

RE: We truly appreciate your recommendation. Unfortunately, due to logistical restrictions we were able to access this test which does not have yet a BAU conversion factor. Nevertheless, the ratios and relationships found in AU/ml with this test correlate with BAU/ml 

• Lines 224-226: Numbers in the text (RR 1.13, 95% CI 1.02-1.26) do not match numbers in Table 2 (female gender, adjusted model; RR 1.12, 95% CI 1.01-1.25)

RE: Thank you for this observation, this was a typo. We have proceeded to correct the numbers in the text (lines 225-226) according to the information provided in Table 2.

• Lines 243-245/Table 3: The higher baseline titers in older individuals are not unexpected as the majority of elderly individuals had received primary immunizations with the much more immunogenic BNT162b2 vaccine. This should be pointed out here. Because the BNT162b2 is overall more immunogenic than the BBIBP-CorV vaccine, it would be helpful to have Table 3 (or an additional table) be split up into the different primary vaccines as well.

RE: Thank you for pointing this out. We have added this information in lines 244-246 as “A possible explanation was that elderly were immunized with the BNT162b2 vaccine which has demonstrated to be more immunogenic than BBIBP-CorV according to some studies”.

• Lines 338-342: While protection induced by inactivated virions could potentially be wider, the statement in its current form is a bit misleading as the lower protection of inactivated virion vaccines compared to the highly immunogenic mRNA vaccines seems well established.

RE: Thank you for your comment, we have proceeded to clarify this concept in line 345 as “In the particular case of inactivated COVID-19 vaccines such as BBIBP-CorV, as these contain additional SARS-CoV-2 proteins such as the nucleoprotein, could produce a wider immunological response than produced by the spike protein”. We were referring to the immune response and not to the protection of the BBIBP-CorV vaccine; we have also specified that it is a theoretical possibility rather than an established fact.

• Lines 349-351: The authors state that antibody titers before the booster inversely correlated with the post-boost titers, suggesting that shorter boosting intervals may not be beneficial. In this study, this observation is very likely to be confounded by the fact that young individuals, which generally respond to the BNT162b2 vaccine with higher IgG titers than older individuals, had received the much less immunogenic BBIBP-CorV vaccine first. This results in the relatively strong increase in younger individuals (poorly immunogenic first vaccine, very strong response to booster). In contrast, older individuals had higher baseline titers due to the more immunogenic BNT162b2 vaccine compared to BBIBP-CorV (although titers in younger individuals receiving BNT162b2 would have been even higher), and then responded less strongly to the BNT162b2 booster dose than younger individuals.

RE: Thank you for your comment, we have deleted the sentence about short interval boosting, in line 354. We left the statement just as an observation that antibody titers before the booster inversely correlated with the titters after booster with an mRNA vaccine, as stated by Goel et al.

• Table 1: While i.) the elapsed time between the second vaccine dose and the booster dose as well as ii.) the time between the booster dose and the follow-up visit are described, information of the time between the first and second dose is missing (line 86 says: at least 21 days apart). As prolonged vaccination intervals are known to affect the humoral vaccine response, it would be helpful for the interpretation of the results and comparison of the groups to also include this measure.

RE: Thank you for your comment. We calculated the median time between the first and second dose, but we left those results out of the manuscript to keep the text length under the number of words required. The time between first and second doses ranged between 20 and 68 days. The median time between first and second doses was 21 days (IQR: 21 - 21). We are including this information in lines 86 and 87.

• Table 1: Make clear within the table what groups the p values are comparing.

RE: P values are result of comparing the (BNT162b2 x 3) group versus (BBIBP-CorV x 2 + BNT162b2) group. We have edited Table 1 to clarify the interpretation of the p values.

Figure 2: Instead of the natural logarithm, it appears advisable to display the data in the much more commonly used log10 format, which will help with comparison to other results. In addition, the legend needs to define lines, boxes and whiskers.

RE: Thank you for this suggestion. There is a big change in the use of log10 or ln, for example references(1)(2)use log10, while(3) use ln. There seems to be no consensus or pre-established guideline on which one should be used over the other; however, from a statistical point of view, the results should not differ beyond what is explained by numerical accuracy. We have performed this checking on the data, for instance we have considered to use the natural logarithm. 

• Figure 3: The 95% confidence intervals are a bit difficult to make out and it would be helpful to display them in a more prominent shade of grey.

RE: Thank you for your comment. We have proceeded to increase the intensity of the grey shading to improve visibility.

• Line 65: “13% fold” does not make sense

RE: Thank you for the comment, we have proceeded to delete the word “fold” in lines 65 and 285.

• Line 67: Make clear what “both” is referring to.

RE: Thank you for the comment and we have clarified in line 67 that “both” refers to the two vaccine regimens (heterologous and homologous) assessed along this study.

• Line 144: Typo, should be BNT162b2.

RE: Thank you for the comment and we apologise for the typo. We have proceeded to correct to BNT162b2 in line 144.

• Line 201: Ranged would be a better word than oscillated.

RE: Thank you for the comment, we have proceeded to put “oscillated” instead of “ranged” in line 201. 

• Lines 212-213: Better to write that there were “no statistically significant” differences than just “no” differences.

RE: Thank you for the comment and we have proceeded to add “no statistically significant” in line 212.

• Line 221: Better: nausea than nauseas.

RE: Thank you for the comment, we have proceeded to correct the word as suggested in line 221.

• Line 309: The lower antibody levels could also be expected because of the overall lower immunogenicity of the BBIBP-CorV vaccine compared to BNT162b2.

RE: Thank you for the comment, we have proceeded to add this sentence in lines 311 and 312, and a reference.

• Line 331-332: Heterologous vaccines are common ‘practice’ for several of the diseases (e.g., HIV, HPV). ‘Has been investigated’ might therefore be a better choice of words than ‘has been practiced’.

RE: Thank you for the comment, we have proceeded to replace the word “practiced” with “investigated” in line 335.

• Line 376: While the authors rightfully acknowledge the lack of analyses on neutralizing serum titers as a limitation of the study, the term ‘broadly’ for the description of the extend of humoral response measurements seems too wide to me (it is a rather narrow measurement of the humoral response). One could rather write “overall binding reactivity” or something similar.

RE: Thank you for the comment. We have proceeded to add the phrase “by the overall binding reactivity” in line 388.

• Line 395: “In addition, the antibody titers rising trend after the second vaccine dose in our study indicates that subsequent boosters could be spaced and prioritized in certain populations such as elderly and immunosuppressed.” I am not entirely sure what this statement is referring to. Trends after the second vaccine dose (prior to the booster) were not investigated. Should this refer to the higher titers after the booster dose compared to after the second vaccine dose?

RE: Thank you for the comment and we apologise for this typo. We meant to refer to the third dose, not the second dose. We have proceeded to make the correction in line 407.

Reviewer #2: 

- Overall, the manuscript is written with a reasonable standard of English but could do with a thorough grammar edit.

The study is interesting, but it is difficult to draw any major conclusions about the impact of booster vaccines for the following reasons:

RE: Thank you for your comments, we have revised the English grammar along the manuscript.

- It is expected that there would be such an increase in antibody titres early after a booster dose, 2 weeks is too early to assess immunogenicity and durability of response and a later sample would be more meaningful in response interpretation.

RE: In this study we aimed to evaluate immediate immunogenicity and observed the antibody peak. However, we agree with your comment, that is the reason we are following these patients for the next six months in order to assess long term immunogenicity and durability of antibody response.

- Only IgG was measured, not neutralising antibody which may be more accurate in interpreting immune response correlating with protection

RE: Thank you for this comment, in the limitations section of the article in lines 388 and 389, we have mentioned that we did not include neutralizing antibodies response.

- In view of this, is preferable to tone these findings down as unfortunately it may be misleading to say that these responses result in protection and to correlate them to protection. The findings are interesting, but need to be focussed on what can be derived, rather than making assumptions

Re: We agree with you that the word “protection” is misleading, so we have proceeded to clarify this concept along the text. In the discussion, we tried to convene the idea that inactivated virions could theoretically produce a wider set of antibodies, not that they could produce a better response or protection. Therefore, we have modified several lines toning down phrases regarding protection and possible vaccine effectiveness, focusing just on immunogenicity. -There are also substantial differences between the homologous and heterologous groups and a comparison is very difficult even with some parameters controlled for

RE: Thank you for your comment. We are aware that the homologous and heterologous groups have pronounced differences, particularly the median age of participants, and the time between boosting and second dose. However, this is the best available data to observe immunogenic response to boosting between groups that were primed differently. Differences in age and time to boosting come from the way the vaccine program rolled up in Peru, and therefore it was almost impossible to have matching comparable groups. In order to control this, we used the best adjustment strategies for observational studies; however, we recognized that residual confounding is a real possibility. We have added this limitation in lines 385-388 as “another important limitation is that due to the vaccine program rolled up in Peru, there were pronounced differences between participants characteristics vaccinated with the homologous and the heterologous regimen, particularly the median age, however, we used the best adjustment strategies for observational studies”.

In addition, all observational studies have an inherent risk of confounding which can be reduced by regression adjustment strategies. Our study adjusted for known confounding variables such as age, gender, comorbidities, prior COVID-19 infection, time until booster, time between samples and baseline IgG levels, this is mentioned in lines 327 and 328. However, the risk of residual confounding will always be present in any observational study.

• In addition:

- How was fever assessed?

RE: Thank you for the comment. We assessed fever by asking participants if they had an oral temperature of 38°C or above. We have added to the manuscript (lines 143 and 144) that safety assessment was self-reported and adverse reactions were inquired during the two-week follow-up visit

-Was any formal measurement of local reactogenicity conducted (measure size of swelling, erythema etc.) Was FDA guidance used to evaluate this?

RE: Reactogenicity was self-reported, as mentioned in lines 143 and 144. We used the Common Terminology Criteria for Adverse Events (CTCAE) as guidance to evaluate reactogenicity. 

- How was prior COVID-19 assessed, given that it is expected that by Dec 21 close to 80% of the global population had been infected with COVID-19 as evidenced by antibody testing. Even asymptomatic infection would give an Ab response and this is likely to confound baseline data. 

RE: We assessed prior COVID-19 and/or documented SARS-CoV-2 infection using a composite definition based on self-reporting and having a prior positive antigenic or molecular test available in the Integrated COVID-19 Register of antigenic and molecular tests (SISCOVID) from the Ministry of health. Asymptomatic infections with no registry of diagnostic tests were not accounted in the study. We are adding this as a limitation: “A further limitation is the under-reporting of prior SARS-CoV-2 due to presence of asymptomatic infections with no testing” in lines 380 – 382.

- In line 225 heterologous has been duplicated- I'm assuming that homologous is the correct term.

RE: Thank you for the comment, “homologous” is the correct term, and it was edited accordingly in the line 277. 

REFERENCES

1. Ligumsky H, Safadi E, Etan T, Vaknin N, Waller M, Croll A, et al. Immunogenicity and Safety of the BNT162b2 mRNA COVID-19 Vaccine Among Actively Treated Cancer Patients. JNCI J Natl Cancer Inst. 1 de febrero de 2022;114(2):203-9. 

2. Pollock KM, Cheeseman HM, Szubert AJ, Libri V, Boffito M, Owen D, et al. Safety and immunogenicity of a self-amplifying RNA vaccine against COVID-19: COVAC1, a phase I, dose-ranging trial. eClinicalMedicine. 1 de febrero de 2022;44:101262. 

3. Haranaka M, Baber J, Ogama Y, Yamaji M, Aizawa M, Kogawara O, et al. A randomized study to evaluate safety and immunogenicity of the BNT162b2 COVID-19 vaccine in healthy Japanese adults. Nat Commun. 14 de diciembre de 2021;12(1):7105.

Regarding Journal requirements:

Point 1. We have reviewed the style requirements as requested.

Point 2. We are including captions for suporting information at the end of the manuscript.

Point 3. We have included the Funder statement in the revised cover letter and an ethics committee contact number 

Point 4. We have uploaded the data base as a Supplementary file

Point 5. We are including captions for suporting information at the end of the manuscript.

---

## [Decision Letter · Decision Letter 1]

15 Sep 2022

PONE-D-22-12508R1Immunogenicity and reactogenicity of a third dose of BNT162b2 vaccine for COVID-19 after a primary regimen with BBIBP-CorV or BNT162b2 vaccines in Lima, Peru.PLOS ONE

Dear Dr. Natalia Gladys Gladys Vargas Herrera,

Thank you for submitting your manuscript to PLOS ONE. After careful consideration, we feel that it has merit but does not fully meet PLOS ONE’s publication criteria as it currently stands. Therefore, we invite you to submit a revised version of the manuscript that addresses the points raised during the review process.

We look forward to receiving your revised manuscript.

Kind regards,

Daniela Flavia Hozbor

Academic Editor

PLOS ONE

Journal Requirements:

Reviewers' comments:

Reviewer's Responses to Questions

**Comments to the Author**

1. If the authors have adequately addressed your comments raised in a previous round of review and you feel that this manuscript is now acceptable for publication, you may indicate that here to bypass the “Comments to the Author” section, enter your conflict of interest statement in the “Confidential to Editor” section, and submit your "Accept" recommendation.

Reviewer #1: (No Response)

Reviewer #2: All comments have been addressed

2. Is the manuscript technically sound, and do the data support the conclusions?

Reviewer #1: No

Reviewer #2: Yes

3. Has the statistical analysis been performed appropriately and rigorously? 

Reviewer #1: Yes

Reviewer #2: Yes

4. Have the authors made all data underlying the findings in their manuscript fully available?

Reviewer #1: Yes

Reviewer #2: Yes

5. Is the manuscript presented in an intelligible fashion and written in standard English?

Reviewer #1: Yes

Reviewer #2: Yes

6. Review Comments to the Author

Reviewer #1: In the revised version of their manuscript, Herrera and colleagues have adequately addressed most of my comments.

A few remaining aspects:

1)

Figure 2 legend is still missing a definition of whiskers, bars, etc.

2)

Given the differences in immunogenicity between the different vaccines, I would still consider a version of Table 3 separated by the primary vaccination regimen informative (see previous comment on “Lines 243-245/Table 3”). Such a table could go into the supplement.

3)

Lines 351-351 (and previous comment on “Lines 349-351”): While the authors have modified their statement in response to my previous comment, I still consider it misleading. The cited work by Goel has determined that the post-boost “fold-change” in neutralization titer (but not the titer itself) inversely correlated with pre-boost titers. Because the comparison of pre- and post-boost titers in this study is strongly influence by the differences in the cohorts (in terms of baseline vaccine and their typical response to mRNA vaccines – see previous comment), I would suggest removing this paragraph.

4)

Lines 84-85: The addition of the information on the time between doses is appreciated. However, it would be placed better in the results section when the two cohorts described. Moreover, differentiation by the primary vaccine type (BBIBP-CorV or BNT162b2) would be informative.

5)

Line 365: This statement could be worded a bit more careful (“may be” or “can be” “explained” rather than “is” explained).

6)

Line 385: Using the term “best” for the description of the used adjustment strategies should be avoided or explained (why are the methods used the ‘best’)?

Reviewer #2: (No Response)

7. PLOS authors have the option to publish the peer review history of their article (what does this mean?). If published, this will include your full peer review and any attached files.

Reviewer #1: No

Reviewer #2: No

---

## [Author Response · Author response to Decision Letter 1]

21 Sep 2022

AUTHOR´S RESPONSE TO COMMENTS:

Reviewer #1: 

• Figure 2 legend is still missing a definition of whiskers, bars, etc.

RE: Thank you for your comment, we have added the definitions in the legend and we have enhanced the Figure 2 resolution. 

• Given the differences in immunogenicity between the different vaccines, I would still consider a version of Table 3 separated by the primary vaccination regimen informative (see previous comment on “Lines 243-245/Table 3”). Such a table could go into the supplement.

RE: Thank you for your comment, we have added a separated table as suggested in the supplement section (Supplementary table 3). 

• Lines 351-351 (and previous comment on “Lines 349-351”): While the authors have modified their statement in response to my previous comment, I still consider it misleading. The cited work by Goel has determined that the post-boost “fold-change” in neutralization titer (but not the titer itself) inversely correlated with pre-boost titers. Because the comparison of pre- and post-boost titers in this study is strongly influence by the differences in the cohorts (in terms of baseline vaccine and their typical response to mRNA vaccines – see previous comment), I would suggest removing this paragraph.

RE: Thank you for your comment, we have removed this sentence as suggested.

• Lines 84-85: The addition of the information on the time between doses is appreciated. However, it would be placed better in the results section when the two cohorts described. Moreover, differentiation by the primary vaccine type (BBIBP-CorV or BNT162b2) would be informative.

RE: Thank you for your comment, we have moved this information to the results section in lines 199 – 200 and also added the time between doses differentiated between the heterologous and homologous vaccine regimens in lines 206 - 208. 

• Line 365: This statement could be worded a bit more careful (“may be” or “can be” “explained” rather than “is” explained).

RE: Thank you for your comment, we have modified the word “is” to “may be” in line 365 as suggested

• Line 385: Using the term “best” for the description of the used adjustment strategies should be avoided or explained (why are the methods used the ‘best’)?

RE: Thank you for your comment, we have replaced the term “best” by the term “robust” in line 385.

---

## [Editor Report · Decision Letter 2]

26 Sep 2022

Immunogenicity and reactogenicity of a third dose of BNT162b2 vaccine for COVID-19 after a primary regimen with BBIBP-CorV or BNT162b2 vaccines in Lima, Peru.

PONE-D-22-12508R2

Dear Dr. Natalia Gladys Gladys Vargas Herrera,

We’re pleased to inform you that your manuscript has been judged scientifically suitable for publication and will be formally accepted for publication once it meets all outstanding technical requirements.

Kind regards,

Daniela Flavia Hozbor

Academic Editor

PLOS ONE
---

## [Editor Report · Acceptance letter]

6 Oct 2022

PONE-D-22-12508R2 

Immunogenicity and reactogenicity of a third dose of BNT162b2 vaccine for COVID-19 after a primary regimen with BBIBP-CorV or BNT162b2 vaccines in Lima, Peru. 

Dear Dr. Vargas Herrera:

I'm pleased to inform you that your manuscript has been deemed suitable for publication in PLOS ONE. Congratulations! Your manuscript is now with our production department. 

Kind regards, 

on behalf of

Dr. Daniela Flavia Hozbor 

Academic Editor

PLOS ONE